# Combined Oral Contraceptives and the Risk of Thrombosis

**DOI:** 10.3390/ijms262211010

**Published:** 2025-11-14

**Authors:** Jamilya Khizroeva, Victoria Bitsadze, Gennady Sukhikh, Maria Tretyakova, Jean-Christophe Gris, Ismail Elalamy, Grigoris Gerotziafas, Daredzhan Kapanadze, Margaret Kvaratskheliia, Alena Tatarintseva, Azaliia Khisamieva, Ivan Hovancev, Fidan Yakubova, Alexander Makatsariya

**Affiliations:** 1Department of Obstetrics, Gynecology and Perinatal Medicine, The I.M. Sechenov First Moscow State Medical University (Sechenov University), 8-2 Trubetskaya St., 119991 Moscow, Russia; vikabits@mail.ru (V.B.); tretyakova777@yandex.ru (M.T.); jean.christophe.gris@chu-nimes.fr (J.-C.G.); ismail.elalamy@aphp.fr (I.E.); grigorios.gerotziafas@inserm.fr (G.G.); margaret.kv@mail.ru (M.K.); tatarintseva_a_y@student.sechenov.ru (A.T.); khisam2003@mail.ru (A.K.); khovantsev_i_v@student.sechenov.ru (I.H.); fi_dan_2017@mail.ru (F.Y.); gemostasis@mail.ru (A.M.); 2FSBI «National Medical Research Center for Obstetrics, Gynecology and Perinatology Named After Academician V.I.Kulakov», Ministry of Health of the Russian Federation, 4, Oparina Street, 117997 Moscow, Russia; g_sukhikh@oparina4.ru; 3Faculty of Pharmaceutical and Biological Sciences, Montpellier University, 34093 Montpellier, France; 4Department of Hematology, CHU Nîmes, Montpellier University, 30029 Nîmes, France; 5Hematology Department, Faculté Privee de Médecine de Marrakech (FPMM), Route Amizmiz, Marrakech 42312, Morocco; 6Hopital Americain de Paris, 55 rue du Châtea, 92200 Neuilly-Sur-Seine, France; 7Sorbonne University, INSERM UMRS-938, Team “Cancer Vessels, Biology and Therapeutics”, Group “Cancer-Hemostasis-Angiogenesis”, Institut Universitaire de Cancérologie, Consultation Thrombosis in Oncology (COTHON), Tenon-Saint Antoine Hospital, AP-HP, 75020 Paris, France; 8Thrombosis Center, Tenon—Saint Antoine University Hospital, Hôpitaux Universitaires Est Parisien, Assitance Publique Hôpitaix de Paris (AP-HP), 4 Rue de la Chine, 75020 Paris, France; 9Center of Pathology of Pregnancy and Hemostasis «Medlabi», 340112 Tbilisi, Georgia; medlabimedlabi@gmail.com

**Keywords:** combined oral contraceptives, thrombosis, composition of COCs, thrombotic effects of COCs, risk factors of thrombosis

## Abstract

Combined oral contraceptives (COCs) remain one of the most popular reversible contraceptive methods worldwide. Still, regardless of the drug composition and duration of therapy, almost all COCs are associated with the risk of venous thrombosis. This review highlights the main pathogenetic mechanisms of thrombosis development during oral contraceptive use. Increase the production of certain clotting factors; a decrease in antithrombin and protein S levels; acquired resistance to activated protein C; a reduction in tissue factor pathway inhibitor (TFPI); indirect endothelial activation; inhibition of endogenous fibrinolysis; regulation of tissue factor by estradiol-sensitive microRNA; homocysteine imbalance caused by decreased intestinal reabsorption of folates and vitamin B-12; reduced bioavailability of nitric oxide (NO) due to high homocysteine levels; higher blood pressure, water retention, insulin resistance, increased levels of pro-inflammatory C-reactive protein (CRP) and uric acid, and antifibrinolytic (plasminogen activator inhibitor 1 type, PAI-1) biomarkers as consequences of NO deficiency; increased platelet adhesiveness and ADP-induced aggregation, which promote fibrinogen binding; and increased expression of pro-inflammatory cytokines are the main thrombotic effects of COCs use. Clinicians should carefully evaluate each patient’s individual risk factors when prescribing COCs and conduct regular monitoring to reduce the risk of complications.

## 1. Introduction

Contraceptive pills remain one of the most popular reversible contraceptive methods worldwide, but many factors need to be considered and managed before selecting a method. Clinicians should carefully evaluate each patient’s individual risk factors when prescribing combined oral contraceptives (COCs) and conduct regular monitoring to reduce the risk of complications. Some studies [1] have shown a three- to seven-fold increase in venous [2] and arterial thrombosis [3]. The risk of VTE should be regularly re-evaluated, as risk factors for thrombosis can change over time. Educating women about the potential risks and benefits of hormonal contraception will help them make informed decisions about their health.

This article is a narrative review aimed at summarizing the current evidence on the mechanisms and risk of thrombosis associated with combined oral contraceptive (COC) use, emphasizing differences between formulations and identifying the safest hormonal combinations in terms of thrombotic risk.

### 1.1. Historical Background

The history of oral contraceptives (OC) dates to the mid-20th century [4]. In the 1950s, American biologists Margaret Sanger and Catherine McCormick funded research on hormonal contraception. Gregory Pincus, along with Ming-Chue Chang, contributed to the basic concept of the pill. In 1956, the first clinical trials of a hormonal contraceptive containing progestin and estrogen were successfully conducted in Puerto Rico. In 1960, the first oral contraceptive, Enovid 10, produced by G.D. Searle & Company, was approved in the United States.

There are four available forms of estrogens: ethinyl estradiol (EE), 17β-estradiol (E2), estradiol valerate (E2V), and mestranol. The estrogen used in the first oral contraceptive was mestranol, which is rarely used today, and the progesterone component was norethynodrel. This medication contained a very high dosage of hormones (9.85 mg of norethynodrel and 150 μg of mestranol), leading to serious side effects, but it marked a revolution in reproductive health.

Nowadays, OC pills contain much lower doses of hormones—ranging from 0.1 to 3.0 mg of progestins and 20 to 50 μg of estrogens [4]. The most common estrogen component in most modern COCs today is EE, and the amount in the tablet determines the estrogenic effects and the risk of thrombosis. One of the goals of reducing the estrogen dose in COCs was to lower the risk of developing thrombotic complications since the risk of thrombosis with estrogen-containing compounds increases as the estrogen dose goes up [5]. Besides reducing the dose of ethinyl estradiol (EE), some combined oral contraceptives (COCs) now include 17β-estradiol, along with various progesterones at different doses. Progestins are the primary component of hormonal contraceptives because they prevent ovulation through their antigonadotropic effects.

### 1.2. Composition of COCs

There is currently no universal and standardized classification for OCs, but hormonal contraceptive drugs can be grouped based on their progesterone composition (Table 1).

First-generation preparations, which are currently rarely used, include contraceptives containing Lynestrol. The second generation of contraceptives, which were released in the 1970s, had a reduced estrogen dosage, and the introduction of levonorgestrel (LNG) lowered the risk of thrombosis. The second-generation progestins (levonorgestrel and norgestrel) have more profound progestational and androgenic effects in combination with lower EE and strong antiestrogen effects. Nevertheless, the androgenic effects of these combined oral contraceptives (COCs) with a progestin component remained evident [6]. Thus, attempts to reduce the unwanted androgenic impacts of COCs led to the development of third-generation contraceptives, which were introduced in the 1980s and included new progestins such as norgestimate, desogestrel, and gestodene. Despite the modification, most progestins in all three generations are testosterone-derived and, to varying degrees, exhibit undesirable androgenic effects. Finally, the fourth generation of progestins, which contains drospirenone and dienogest (DNG) was designed to be closer in activity to endogenous progesterone than previous generations of progestins and is associated with the lowest risk of side effects.

Oral contraceptives quickly gained popularity despite initial controversies and legal restrictions. Over time, contraceptive formulas improved, with reduced hormone dosages significantly lowering the risk of side effects and increasing the safety of these medications. Today, oral contraceptives are widely used to prevent unwanted pregnancies and to treat various gynecological disorders. Despite the potential risks mentioned earlier, it is essential to recognize the many beneficial effects of hormonal contraceptives. These include a decreased risk of ovarian and endometrial cancer, regulation of the menstrual cycle, and relief from premenstrual syndrome symptoms [7]. However, increasing reports of thromboembolic events since the first use of combined oral contraceptives (COCs) and those associated with COCs have raised concerns about a possible causal relationship between hormonal therapy and the risk of thrombosis [8].

## 2. Methodology

A structured literature search was performed across PubMed, Scopus, and Web of Science databases to find studies published from January 2010 to June 2024. Various combinations of MeSH terms and keywords were used, including: “combined oral contraceptives,” “venous thromboembolism,” “thrombosis,” “risk factors,” “progestins,” and “estetrol drospirenone.” Preference was given to systematic reviews, meta-analyses, and clinical studies written in English. Reference lists of relevant publications were also screened to find additional sources.

## 3. Pathogenesis of Thrombotic Complications with Hormonal Contraceptives

Pathogenetic mechanisms of thrombosis development during oral contraceptive use are very diverse but are still not fully understood. The thrombotic effects of COCs are probably mediated through their effects on the blood coagulation system (Figure 1) [9].

### 3.1. Increase the Production of Some Clotting Factors

The rise in coagulation factors related to OCs may be one of the mechanisms that increases the risk of cardiovascular events in contraceptive users. There is a strong link between OC use, especially with higher estrogen doses, and the risk of thromboembolism. Estrogens in OCs increase the synthesis and activity of clotting factors such as fibrinogen (FI), prothrombin (FII), and factors VII, VIII, and X, while decreasing factor V (FV), which promotes hypercoagulability. It is now well recognized that factor V has both pro- and anticoagulant properties [10]. Since FV acts as a cofactor with protein S and effectively inhibits FVIIIa activity, a reduction in FV levels may contribute to the thrombotic effects of OC [11]. The increase in FII and FVII, along with a moderate decrease in FV, is associated with a significantly higher risk of thrombosis. These changes depend on the estrogen dose, the mode of estrogen delivery [12], and the type of progestogen used in COCs [13]. The effect is more pronounced with the desogestrel-containing OC (third generation) compared to the levonorgestrel-containing OC (second generation) [14].

### 3.2. Decrease in the Activity of Natural Anticoagulants

At the same time, there is a decrease in the activity of natural anticoagulants, such as protein S and antithrombin (AT), along with reduced sensitivity to activated protein C (APC), which also enhances the blood’s clotting properties.

#### 3.2.1. Decreased Antithrombin Levels

Antithrombin (SERPINC 1) is a liver-produced glycoprotein that belongs to a family of serine protease inhibitors and is actively involved in the coagulation cascade. AT binds to serine factor II (thrombin), factor IXa, and factor Xa, which inhibit the blood clotting process. Several studies have established a relationship between oral contraceptive use and changes in AT level [15,16]. Burkman and colleagues found that antithrombin levels in different evaluated oral contraceptive groups decreased from baseline by 19.7% and 28.8% for the immunologic and activity methods, respectively. Initially, they suggested that the estrogen component in OCs is responsible for changes in antithrombin. However, it was later revealed that the highest decline in antithrombin level was observed in type O blood group COC users (31.6%) taking high doses (50 μg) of ethinyl estradiol, and in non-type O women (38.9%) taking the lowest progestin dose (0.5 mg norethindrone) [15,16].

#### 3.2.2. Acquired Resistance to Activated Protein C (APC-R)

The protein C system is the natural anticoagulant system, regulating coagulation, maintaining blood fluidity, preventing thrombosis, and thereby preventing vascular injury and stress. The key protease of the protein C system is activated protein C (APC). Normally, APC inhibits the coagulation cascade by cleaving peptide bonds in FV/Va and VIII/VIIIa. Circulating inactive factor V can exhibit either procoagulant or anticoagulant activity, depending on modification by pro- or anticoagulant enzymes. Under the influence of thrombin, the active form of factor V is produced, which has procoagulant activity. After proteolytic inactivation by activated protein C, FVa is degraded into inactive factor FVi, which effectively downregulates the coagulation and limits blood clot formation. In addition to its anticoagulant activity, APC exhibits cytoprotective and anti-inflammatory effects on vascular endothelial cells, neuronal cells, and various immune system cells. The most common cause of the inability of APC to cleave factor Va and/or factor VIIIa (APC-resistance) is the factor V Leiden mutation.

The use of COCs may result in the development of resistance to the anticoagulant action of APC, regardless of the presence of a genetic mutation of factor V Leiden, especially in women using third-generation (desogestrel-containing) COCs compared with users of second-generation (levonorgestrel-containing) oral contraceptives [10].

However, some studies indicate that COC-induced APC resistance (APC-R) has a low prothrombotic potential and that COC-associated thrombosis occurs rarely, particularly in comparison to high-risk situations, such as major surgery [17]. At the same time, the authors of this study suggest that thrombin generation may increase during COC use when additional risk factors are present.

Protein S is a natural anticoagulant that acts as a cofactor for activated protein C and, in addition, exhibits anticoagulant properties by directly inhibiting thrombin formation (APC-independent action). Protein S (total and free) is remarkably reduced when taking OCs containing 3rd generation progesterone and in postmenopausal women [18].

Most users of 3rd generation contraceptive pills have acquired resistance to APC, which is mediated not only by a decrease in PS but is also associated with changes in the level of sex hormone-binding globulin (SHBG), a liver protein, whose synthesis is highly estrogen dose-dependent [19]. Oral administration of EE alone causes a significant dose-dependent increase in SHBG. In contrast, administration of progestogens results in a decrease in SHBG, the extent of which depends on the androgenic activity of the progestogen [20]. Additionally, HSPG can serve as a measure of total estrogenicity and an indirect marker of thrombosis in OC users [21]. When taking OC with high concentrations of estrogens, there is an increase in the level of SHBG. At the same time, it decreases depending on the antiestrogenic properties of the progesterone component of pills [22]. For example, there is a 50% SHBG increase with LNG-containing contraceptives, 150% with norgestimate, 250–300% with drospirenone and dienogest, 200–300% with desogestrel and gestodene, and 300–400% with cyproterone acetate, carrying a higher risk of VTE [20,21,22]. OCs containing drospirenone, dienogest, cyproterone acetate, and norgestimate may cause more pronounced alterations to APC and, therefore, lead to a higher risk of VTE compared to users of 2nd-generation OCs.

#### 3.2.3. Decrease in TFPI Levels

The anticoagulant properties of endothelial cells (ECs) predominate over procoagulant properties due to the expression of proteins such as tissue factor pathway inhibitor (TFPI), which is the primary natural inhibitor of TF-induced coagulation. TFPI inhibits the FVIIa/tissue factor complex, thereby blocking the initiation of the coagulation process. In 2003, Dahm and colleagues assessed the TFPI level (total and free) and its activity in COC users compared to non-using women [23]. Additionally, they investigated whether a decreased level of TFPI is a risk factor for the development of deep vein thrombosis (DVT). The results showed that forty-one percent of oral contraceptive users had a more profound decrease in free and total TFPI compared with premenopausal nonusers, who had lower levels than men and women in post menopause [23]. They revealed that plasma TFPI level below the 10th percentile is a weak risk factor for DVT (10th percentile = 9 ng/mL), and further analyses found that lower TFPI levels (e.g., cut-off at the 5th and 2nd percentiles) were associated with a higher odds ratio for DVT. The authors concluded that this relationship between higher risk of VTE and lower TFPI levels suggests a threshold level for the protective effect of TFPI against DVT.

### 3.3. Effects on Endothelial Cells

Both estrogen and progesterone receptors are present on vascular endothelial cells and smooth muscle cells, suggesting that these compounds may influence vascular function [24]. It has been well established that 17β-estradiol, the naturally produced estrogen, exerts protective cardiovascular effects through interaction with multiple cell types via estrogen receptors (ER), including nuclear estrogen receptors ERα and ERβ, membrane ER, and G protein-coupled estrogen receptor. ECs are the primary target of E2’s beneficial effects, such as neovascularization, which helps reduce ischemic injury.

To identify whether COC hormones provoke endothelial prothrombotic activity, Bouck et al. conducted a study using an in vitro model of the plasma-endothelial interface [25]. They treated the human umbilical vein and microvascular endothelium with ethinylestradiol and/or drospirenone. However, neither of these drugs alone nor their combination resulted in changes in the gene expression encoding anti- or procoagulant proteins (TFPI, THBD, F3), integrins (ITGAV, ITGB3), or fibrinolytic mediators (SERPINE1, PLAT). Also, there was no increase in thrombin or fibrin generation. Thus, the authors claim that there is no direct procoagulant effect of COCs (EE and drospirenone) on endothelial cells. However, it is generally appreciated that EE carries a dose-dependent risk of VTE, and drospirenone increases VTE risk 6.3-fold compared with no OC users and 2–3-fold compared with levonorgestrel [25]. The authors provide possible explanations for the contradictory results obtained. First, women with VTE may carry undetected risk factors or polymorphisms that interact with OCs, promoting endothelial dysfunction and thrombus formation. Secondly, the results do not exclude an indirect role of endothelial activation in OC-associated VTE (activation of hepatocytes, platelets, neutrophils, monocytes, or hormone-responsive tissues), which secondarily release factors that activate ECs and trigger procoagulant activity on the vessel wall. Therefore, it is necessary to continue the search for possible mechanisms of OC effects on the endothelium [25].

### 3.4. Effects on the Fibrinolytic System

Although the procoagulant effects of estrogens are well documented, their impact on the fibrinolytic system has only recently emerged as a topic of interest. OCs significantly affect the fibrinolytic process, which may also lead to thrombosis formation. Fibrinolysis plays a key role in maintaining the balance between clot formation and destruction by preventing excessive fibrin accumulation in blood vessels. The main components of this system include plasminogen, plasminogen activators, and inhibitors, such as tissue plasminogen activator (tPA), and plasminogen activator inhibitor-1 (PAI-1) [26]. During OC use, the levels of tPA activity, plasminogen, plasmin-alpha2-antiplasmin complexes, and D-dimer increase significantly (by 30–80%), while the levels of PAI-1 antigen, PAI-1 activity, and tPA antigen decrease (by 25–50%), suggesting the profibrinolytic effect of OC [27]. However, these changes were counteracted by higher levels of the thrombin-activatable fibrinolysis inhibitor (TAFI), resulting in a net prothrombotic state. Inhibition of endogenous fibrinolysis occurs independently of factor XI and is more pronounced with desogestrel-containing OC, rather than levonorgestrel-containing OC [27].

### 3.5. Regulator of Tissue Factor Through miRNA

miRNAs are small, single-stranded, non-coding RNAs that target mRNAs via the 3′-UTR and regulate target gene expression at a post-transcriptional level by RNA silencing, leading to translation inhibition and potential mRNA degradation [28]. Almost every biological process in the body is mediated by microRNA. Several epidemiological studies have identified microRNAs that can modulate the expression of target mRNAs associated with thrombosis and hemostasis and may serve as potential biomarkers for VTE in the general population. Through interaction with the 3′-UTR, miRNAs directly regulate multiple hemostatic functions and are involved in the relationship between the hemostatic system and inflammation, controlling the expression of fibrinogen, coagulation factors III, VII, VIII, IX, XI, von Willebrand Factor (vWF), PAI-1, antithrombin, proteins C, and S.

Tian et al. identified an estradiol-sensitive microRNA (miR), miR-494-3p, which suppresses protein S expression and hypothesized that miRs may similarly regulate other coagulation factors. They found a direct interaction between miR-365a-3p and the 3′-untranslated region of tissue factor mRNA. Tissue factor (TF), as the primary trigger of blood coagulation, was upregulated by estradiol treatment in plasma samples obtained from pregnant women and women on the contraceptive pill compared with sex-matched controls. Thus, miR-365a-3p can be considered a new regulator of tissue factor that initiates thrombin formation [28].

### 3.6. Effects on Nitric Oxide (NO) and Homocysteine Levels

Possible blood clot formation pathways during OC use include effects on nitric oxide (NO) and homocysteine levels. NO is an important biomolecule that plays a key role in the regulation of hemostasis by influencing various components of the blood coagulation system. It is synthesized by endothelial cells via the enzyme nitric oxide synthase (eNOS) and acts as a vasodilator, antiaggregant, and anticoagulant [29]. Many of the NO effects are cardioprotective. Endogenous 17beta-estradiol has a direct vasodilator effect, increasing NO synthesis in endothelial cells and platelets [30]. Numerous studies were conducted to assess the impact of exogenous female hormones on NO levels. In the study by Olatunji, it has been shown that COC-treated female rats (1.0 μg ethinylestradiol + 5.0 μg levonorgestrel) with NO deficiency had significantly higher blood pressure, water retention, insulin resistance, increased levels of pro-inflammatory (C-reactive protein; CRP and uric acid) and antifibrinolytic (PAI-1) biomarkers [31].

In turn, homocysteine reduces the bioavailability of NO. Homocysteine imbalance leads to redox imbalance, increased protein/nucleic acid/carbohydrate oxidation, lipid peroxidation, and ultimately cytotoxicity [32]. One mechanism for triggering the coagulation cascade is the denudation of the subendothelial matrix and endothelial smooth muscle cells by homocysteine metabolism products. It is widely accepted that hyperhomocysteinemia is an independent risk factor for cardiovascular diseases [32]. Detoxification of homocysteine occurs through remethylation to methionine (B vitamins and folate-dependent and independent routes) and transsulfuration to cystathionine, which requires the activity of essential transsulfuration enzymes (cystathionine β-synthase, cystathionine γ-lyase). Homocysteine is expressed in all tissues, but the brain lacks cystathionine γ-lyase and some other enzymes, being dependent on folic acid/B12 pathways of homocysteine disposal and making the brain parenchyma vulnerable to excess homocysteine.

Moreover, hormonal contraceptives reduce intestinal reabsorption of folates and vitamin B-12, which leads to a significant decrease in their levels in women taking OCs, even when their intake with food is controlled, and can lead to hyperhomocysteinemia and megaloblastic anemia [33].

An additional pathway is the induction of inflammatory responses by homocysteine, which increases proinflammatory cytokines and subsequently induces cellular apoptosis [34]. Fallah et al. confirmed the effect of OCs on homocysteine and NO concentrations, as well as their role in the development of a prothrombotic state [33]. They found that COCs increase homocysteine levels and decrease NO levels, which may result in increased cardiovascular risk [33]. However, the studies of Merki-Feld et al. [35] and Momeni et al. [36] demonstrated no changes in NO and homocysteine levels in women taking OCs, which requires further studies of this mechanism in the development of thrombosis.

Anyway, the alteration in homocysteine levels and inhibition of NO induced by COCs could suggest that the use of oral contraceptives should be reviewed in women with increased cardiovascular risk factors.

### 3.7. COCs and Platelets

Platelets are actively involved in the coagulation processes and play a central role in the development of arterial and venous thrombosis. The effect of OCs on platelet counts and function remains unclear. Estimates of the estrogen effect on platelet counts vary widely in the literature. For example, Fox et al. demonstrated that high doses of estrogen (500 µg/kg) resulted in an increased level of mature CD41+ megakaryocytes and circulating platelets in mice [37]. However, administration of 10–100 μg/kg/day of 17β-estradiol (17β-E2) for 10 days resulted in a decrease in platelet counts.

There are many studies with contradictory results regarding the effects of COCs on platelet function. For example, a survey by Saleh et al. showed no evidence of altered platelet function [38]. They found no significant differences (*p* = 0.05) in platelet aggregation using adenosine diphosphate (ADP) and collagen as inducing agents in women using monophasic COCs (n = 12), triphasic COCs (n = 13), Norplant (n = 19) after 3 months of hormone intake compared to 29 control healthy women. The research group led by Bulur et al. found that the use of oral contraceptives did not cause changes in the mean platelet volume (MPV) value in young women [39]. Beltran et al. performed a comparative study of four different hormonal combinations (a transdermal release system that contains 6 mg of norelgestromin (NGMN)/0.600 mg of EE; a vaginal release system containing 11.7 mg of Etonogestrel/2.7 mg of EE; oral COC containing 2 mg of chlormadinone acetate/0.030 mg of EE; COC with 3 mg of drospirenone/0.030 mg of EE) on platelet count and coagulation parameters after 6 months of intake [40]. A statistically significant increase was seen in platelet count and prothrombin activity in the group treated with NGMN/0.600 mg of EE.

Oral contraceptives increase platelet adhesiveness and ADP-induced aggregation, which causes fibrinogen binding [41]. Moreover, in women on OCs, the platelet lipid biosynthesis is increased dramatically, which implies a higher response of platelets to thrombin-induced aggregation [42]. The lipid membrane is crucial for initiating the coagulation cascade. The procoagulant platelet acts as a scaffold that supports the assembly of the prothrombinase complex, which leads to the explosion of thrombin (so-called thrombin burst), the formation of fibrin, resulting in the formation of a fragile fibrin network [43]. Moreover, procoagulant platelets can be considered a new player in the process of thromboinflammation, performing a proinflammatory role by releasing platelet microparticles and inorganic polyphosphate. Once platelets become activated, their aggregation begins, and von Willebrand factor (vWF) is released from alpha granules of platelets.

Willebrand factor binds platelets when the integrity of the vascular wall is compromised (thereby participating in primary hemostasis) and when it binds to factor VIII. VWF protects FVIII from the proteolytic activity of protein C and delivers it to the site of vascular injury, playing an important role in secondary hemostasis. vWF facilitates the attachment of platelets to the damaged endothelium due to the formation of an adhesive bridge [44]. There have been reports that estrogen directly stimulates the synthesis of vWF by endothelial cells, resulting in a slight increase in endothelial replication [44].

### 3.8. COCs and Inflammation

OCs also play a significant role in inflammatory processes and the immune system. In vitro and in vivo models have demonstrated that low-dose 17β-estradiol (17β-E2) administration is associated with increased expression of pro-inflammatory cytokines [45]. The regulation of inflammatory mechanisms likely occurs through nuclear/membrane estrogen receptor-mediated stimulation of Toll-like receptor 4 ligands, leading to increased IL-1β and IL-6, inhibition of phosphatidylinositol 3-kinase (PI3K) pathway activity, and regulation of NF-κB-mediated transcriptional activity [46,47]. Results of a new UCLA Health study, published in the journal Brain, Behavior, and Immunity [48], found that OC users exhibit numerous signs of chronic inflammation, including elevated C-reactive protein levels and a greater risk of developing mood and autoimmune disorders.

For comparison, it is worth mentioning that endogenous estrogen is negatively associated with C-reactive protein [49]. However, the opposite relationship has been found between exogenously administered female hormones and CRP levels, suggesting that the effect of sex steroid hormones on inflammatory activity in the body may depend, among other factors, on their source. Cauci et al. showed an association between the presence of high-sensitivity C-reactive protein (hsCRP), indicating an ongoing inflammatory process in women of reproductive age taking OCs [50]. Elevated hsCRP levels ≥ 2.0 mg/L are considered a risk factor for cardiovascular disease. These hsCRP concentrations were found in 41.0% of women taking OCs compared to controls [OR = 6.6, 95%CI 3.5–12.4, *p* < 0.001] [50].

### 3.9. Role of Lipoprotein(a) in Thrombosis

Recent studies have also highlighted the potential role of lipoprotein(a) (Lp(a)) as a prothrombotic factor in the context of COC use [51]. Elevated plasma levels of Lp(a) may promote both venous and arterial thrombosis through different pathways, including antifibrinolytic and proinflammatory mechanisms, such as inhibiting plasminogen activation and increasing oxidative stress. Oxidized phospholipids carried by Lp(a), along with vascular endothelial dysfunction, may further raise the thrombotic risk associated with COC use [52]. Although the clinical significance of Lp(a) in women using COCs remains limited, measuring it could provide additional insights for women with hereditary or acquired thrombophilia, supporting a more personalized assessment of thrombotic risk.

Table 2 presents a comprehensive overview of the key pathogenetic mechanisms that link COC use to a prothrombotic state, including both classical and recently discovered molecular pathways such as NET formation and elevated Lp(a) levels.

## 4. Hormonal Contraception and the Risk of Thrombosis

A large meta-analysis and systematic review of 3110 publications (25 publications reporting on 26 studies) was conducted to comprehensively assess the risk of venous thromboembolism in women using different combined oral contraceptives [53]. The results of the study reported that the use of COCs leads to almost a four-fold increase in the risk of venous thrombosis (OR 3.5, 95% CI 2.9–4.3 In comparison, the incidence of VTE in individuals not using contraception was 1.9 and 3.7 per 10,000 women per year [53].

### 4.1. Risk of Venous Thrombosis

Mohammed Al Sheef et al., in their five-year retrospective review, analyzed the risk of various thrombotic complications in 100 female patients who were taking OCs [54]. In this, several types of complications, such as DVT, pulmonary embolism (PE), and other various foci of thrombosis and embolism, were identified [54].

Venous thrombosis is also known to be associated with a high risk of mortality, which also emphasizes the importance of this problem [55]. According to the data, the risk of death from PE in women of reproductive age using OCs is 10.5 (6.2–16.6) per million women per year (95% CI: 6.2–16.6) [56]. In this case, PE developed in 78% of patients taking OCs. At the same time, 10% of patients with VTE also developed PE. DVT was noted in 52% of patients, and most of them had thrombosis in the left leg. Rarer thrombotic complications, according to the study, include cerebral venous thrombosis (n = 12), Budd-Chiari syndrome (n = 2), and visceral thrombosis (n = 6). The estimated age of patients with PE and VTE was 34 ± 8.1 years, with a mean duration of COC intake of 13.6 ± 2.7 months. Patients mostly used COCs such as cyproterone acetate 2 mg + ethinylestradiol 0.035 mg (n = 6), gestodene 075 mg + ethinylestradiol 0.03 mg (n = 18), ethinylestradiol 0.035 mg + desogestrel 0.15 mg (n = 14), and drospirenone 3 mg + ethinylestradiol 0.03 mg (n = 8). Additional risk factors for PE and VTE associated with COCs intake, which have a synergistic effect on the occurrence of thrombosis of various causes, were also identified. Such risk factors include morbid obesity (53%), recent surgical interventions (15%), a family history of VTE (9%), thrombophilia (8%), and diabetes mellitus (5%).

Some reports demonstrated that the risk of DVT (OR = 6.6, 95% CI 5.4–8.0) is slightly higher than that of PE (OR = 3.9, 95% CI 3.2–4.8) [57]. The development of VTE is also directly related to associated risk factors, such as increased body mass index (BMI), age, and the presence of hereditary or acquired thrombophilia [56].

Other side effects that are less life-threatening include arm deep vein thrombosis (OR = 1.9, 95% CI 1.1–3.4) [56,57,58]. Visceral venous thrombosis should also be included among the rare localizations of VTE. The risk of cerebral sinus vein thrombosis (CSVT) increases 7-fold when taking OCs [59]. Thrombosis of deep and superficial cerebral veins, as well as cerebral sinuses, should be included in CVT. There are reports on the occurrence of isolated splenic vein thrombosis (ISVT) with the history of OCs use [60]. ISVT is a rare form of venous thrombosis occurring without pancreatic disease. This condition is characterized by acute abdominal pain and chronic left-sided portal hypertension despite a lack of disease control. In this clinical case, the patient was treated with oral rivaroxaban as an anticoagulant and systemic thrombolysis using urokinase. These pathological conditions are relatively uncommon, and the risk of their development depends not only on the use of OCs but also on the presence of concomitant risk factors.

### 4.2. Risk of Arterial Thrombosis

In addition to venous thrombosis, COC use may increase the risk of arterial thromboses, such as ischemic stroke, by twofold to fourfold [3,61]. However, risk factors for arterial thrombosis differ from those for venous thrombosis. For example, smoking is a risk factor for myocardial infarction associated with OC use but does not significantly affect the risk of venous thrombosis in COC users [62].

The risk of acute ischemic stroke (AIS) associated with high estrogen states arising from exogenous estrogen (COCs, hormone replacement therapy), or in the hyperestrogenic prothrombotic setting of pregnancy and the post-partum period has been well documented [63]. The risk of stroke increases by 2 to 3 times in women with additional risk factors, such as smoking, hypertension, and hypercholesterolemia [63]. The combination of estrogen-progestin hormonal contraception and smoking is associated with an elevated risk of various vascular complications, including stroke and myocardial infarction. These complications are linked to pathological changes in coagulation factors, specifically an increase in the level of the fibrinolysis inhibitor PAI-1 [64].

### 4.3. Duration of Oral Contraceptive Use and Risk of Thrombosis

The time interval from the start of COC use and the development of VTE/PE was an average of 13.6 ± 2.7 months [65]. In 34% of women, thrombosis developed after less than 3 months of taking OCs, in 24.5% after 3–6 months, in 15.1% after 7–12 months, and in 26.4% the interval was more than 12 months.

According to van Hylckama Vlieg’s research, the risk of VTE development was highest in the first three months of OC use (odds ratio, 12.6; 95% CI, 7.1 to 22.4) [66]. After one year of COC intake, the risk of venous thrombosis had decreased to an overall estimate of a five-fold increased risk compared with non-using women.

The risk of thrombosis decreases over time, possibly because high-risk patients stop using hormonal contraceptives due to thrombosis. In particular, the risk of VTE returns to the background 8–12 weeks after the complete discontinuation of hormonal contraceptive preparations [67], but still, it remains higher than in non-users.

### 4.4. Composition of Oral Contraceptives and Risk of Thrombosis

The risk of thrombotic complications depends not only on the duration of drug use and associated risk factors but also on the composition of the OCs. The type of oral contraceptives evaluated in the study by AlSheef was known in 46%. In 6%, it was a combination of cyproterone acetate 2 mg/EE 0.035 mg; in 18%—gestodene 0.075 mg/EE 0.03 mg; in 14%—desogestrel 0.15 mg/EE 0.035 mg; in 8%—drospirenone 3 mg/EE 0.03 mg [54]. Reducing the dose of estrogen from 100 µg to 50 µg in COCs decreased the risk of future thrombotic events, but no evidence that lowering the dose of estrogen by less than 50 µg prevents VTE [54]. However, some data showed that when the estrogen dose was reduced to 30–20 µg, the risk of venous thrombosis was 0.8 (95% CI 0.5–1.2), while for a 50 µg estrogen dose, the risk was slightly higher at 1.9 (95% CI 1.1–3.4) [68]. At the same time, higher estrogen doses are directly linked to an increased risk of thrombosis for OCs containing desogestrel and gestodene. When the ethinylestradiol dose was lowered to 20 µg, OCs with LNG in their formulation showed no significant reduction in venous thrombotic risk complications [69].

For a long time, the increased risk of thrombosis was attributed solely to the estrogen component of oral contraceptives. However, subsequent episodes of thromboembolic events have challenged the concept that reducing the dose of estrogen in hormonal contraceptive pills eliminates the risk of VTE. Since low-dose second- and third-generation COCs contain the same constant dose of estrogen component, the differences in thrombotic risk most likely reflect a progestogen-specific effect, indicating that certain progestins may increase the risk of thrombosis [70]. Although progestogens may have estrogen-like effects, including on hemostasis, there is no data on the effect of levonorgestrel or desogestrel on the protein C pathway in the absence of an estrogen component.

Vlieg et al., in the MEGA (Multiple Environmental and Genetic Assessment of risk factors for venous thrombosis) study, which included 1524 patients with VTE, evaluated the risk of thrombotic complications by focusing on the estrogen dose and progestin type of OCs [66]. According to the study, the risk of venous thrombosis was much higher in women taking commercially available OCs than in those who do not use such hormones (OR 5.0, 95% CI 4.2–5.8). Levonorgestrel-containing OCs use increases the risk of VTE almost fourfold (odds ratio 3.6, 95% CI 2.9–4.6), gestodene was associated with a 5.6-fold increase (95% CI 3.7 to 8.4), desogestrel—7.3-fold (5.3 to 10.0), and cyproterone acetate—6.8-fold (4.7 to 10.0) compared with non-use [66]. The most frequently used type of progesterone was levonorgestrel. To assess the risk associated with progesterone type, the study authors directly compared COCs with different types of progesterone with contraceptives containing levonorgestrel. All contraceptives contained the same dose of estrogen. The relative risk of VTE development in women taking COCs containing 30–35 mcg ethinylestradiol and gestodene (OR 1.6, 95% CI 1.0 to 2.4), desogestrel (2.0, 1.4 to 2.8), cyproterone acetate (2.0, 1.3 to 3.0), or drospirenone (1.7, 0.7 to 3.9) was similar and approximately 50–80% higher than for levonorgestrel-containing OCs. Among 3rd-generation COCs, the risk was slightly higher for desogestrel than gestodene (odds ratio 1.3, 95% CI 0.8 to 2.2). Higher doses of EE were associated with a higher risk of thrombosis. This dose–response relationship was observed for gestodene, desogestrel, and levonorgestrel. Even if most women use monophasic pills, the risk of VTE associated with triphasic OCs (levonorgestrel or gestodene with 30–40 mcg EE) has also been assessed. It has shown an increased risk like that of monophasic COCs compared with non-users [64].

Stegeman et al. demonstrated the most comprehensive information on the risk of VTE using OCs in their meta-analysis [53]. They concluded that regardless of the drug composition and duration of therapy, almost all OCs are associated with the risk of venous thrombosis [53]. There is an association between individual generations of OCs and the risk of venous thrombosis. Thus, first-generation OCs had a 3.2-fold increased risk of thrombotic complications compared with women not taking OCs (95% CI 2.0–5.1). Second-generation OCs were characterized by a 2.8-fold (95% CI 2.0–4.1) increased risk of thrombosis, while third-generation OCs resulted in a 3.8-fold (95% CI 2.7–5.4) increased risk compared with patients not using hormonal contraception. This meta-analysis also gives an example of the most dangerous drug in terms of thrombosis, which is 50LNG. At the same time, 20LNG and 20GSD contributed to venous thrombosis in a smaller percentage of cases. A dose-dependent effect was observed for gestodene, desogestrel, and LNG—the risk of thrombosis increased significantly with increasing OC dosage [68].

OCs containing cyproterone acetate, which also has an anti-ovulatory effect similar to that of progestogens, should be categorized as a distinct group of drugs. Data show that the risk of venous thrombosis in women using OCs with cyproterone acetate and/or drospirenone is approximately 50–80% higher compared to those using LNG-containing OCs [70]. However, the use of OCs with LNG is also associated with a fourfold increased risk of VTE (OR = 3.6:2, 9:4.6) [53] compared to women who do not take OCs. Consequently, the safest option regarding thrombosis risk is LNG, with the risk increasing gradually with gestodene, drospirenone, and cyproterone acetate, respectively. Notably, desogestrel is linked to the highest risk of thrombosis [67]. There is no conclusive, reliable data confirming a difference in thrombotic risk between second- and third-generation OCs [71,72,73].

Other studies also confirm that the risk of thrombotic complications depends on the type of progestogen in OCs [69,70]. According to a literature review by Deeksha Khialani et al., new kinds of OCs containing the estrogenic component E2 are safer in terms of thrombotic risk compared to those with EE. EE has been repeatedly shown to increase the risk of thrombosis in the population [65]. Consequently, new OC formulations have been developed that include estradiol valerate (E2V) and 17β-E2 as estrogen components. 17β-E2 is a natural estrogen with the fewest metabolic side effects [71]. Additionally, E2V pairs well with DNG in a four-phase dosing regimen to offer practical and safe contraception. DNG provides a strong progestational effect, suppressing ovulation and reducing menstrual bleeding. The DNG/E2V combination has fewer cardiovascular adverse effects and is also less likely to cause VTE 0.5 (95% Cl: 0.2–1.5) [71]. Also, 17β-E2 combines well with nomegestrol acetate (NOMAC), providing effective contraception and reducing the duration of menstrual bleeding [72]. NOMAC is probably associated with a lower risk of VTE than other OCs, such as those with LNG and EE in the formula.

In recent years, a new combined oral contraceptive containing estetrol (E4) and drospirenone has been introduced [73]. Estetrol is a natural fetal estrogen with selective action on estrogen receptors and has a weaker effect on hepatic synthesis of coagulation factors compared to ethinylestradiol. Early clinical data and postmarketing observations suggest that E4/drospirenone may have a more favorable hemostatic profile and potentially a lower VTE risk than EE-containing preparations with drospirenone, while still offering effective contraception [74,75]. However, these findings still require confirmation in larger real-world study cohorts.

Table 3 compares the relative venous thromboembolism (VTE) risk associated with various COC formulations by estrogen type and progestin generation, providing a visual framework for understanding how formulation differences affect hemostasis. Formulations containing estetrol/drospirenone are highlighted as emerging options with potentially lower thrombotic risk.

### 4.5. Age-Dependent Risk of Thrombosis

When choosing a particular OC product, each reason for the potential risk of VTE or PE must be balanced against the benefit that a woman may derive from hormonal contraception. The patient’s age seems to play a significant role in the development of thrombotic complications. To assess the age-related risk of VTE, the MEGA study calculated the risk for OC users in different age groups. Data from 1524 women taking COCs who developed thrombosis of various localizations showed that the risk of thrombotic complications increases with age. The relative risk (95% CI) of VTE associated with OC use for women aged ≤30 years was 3.1 (2.2 to 4.6), 30–40 years—5.0 (3.8 to 6.5), and 40–50 years—5.8 (4.6 to 7.3) compared with non-users of OCs [66].

The age-related increase in VTE incidence, from 1.84 per 10,000 woman-years in women aged 15–19 to 6.59 per 10,000 woman-years in women aged 45–49, was confirmed in a Danish National cohort study [76]. The study analyzed 4213 cases of VTE occurring in 3.4 million women per year, of which 2045 cases of thrombosis were among current OC users.

Younger women are more likely to use OCs. In adolescents, OCs may be prescribed for reasons beyond contraception. For instance, they are often used to treat endometriosis and dysmenorrhea, as well as for hormone replacement therapy. However, many adolescents (about 55%) use OCs specifically for effective and safe contraception [77]. The risk of thrombotic complications in adolescents using OCs is 3–5 times higher compared to those not on hormonal therapy. According to 2011 statistics, the annual risk of venous thrombosis in healthy adolescents not taking OCs is 0.05%. The most suitable medications for adolescents are LNG, norgestrel, and norethindrone, which are associated with a lower percentage of venous thrombosis cases. In individuals with thrombophilia, a purely progestin-based contraceptive (PBC) can be used, as PBCs at standard doses generally do not negatively affect hemostasis.

### 4.6. Personal History of VTE and COC Use

In women with a history of venous thromboembolism (VTE), the use of COCs is contraindicated, whether the event was provoked or unprovoked [78]. Estrogen-containing contraceptives considerably increase the risk of recurrent thrombosis by promoting a hypercoagulable state through increased production of clotting factors, decreased natural anticoagulants, and reduced fibrinolytic activity. According to the World Health Organization (WHO) and CDC Medical Eligibility Criteria, COCs are classified as Category 4 (“unacceptable health risk”) in women with previous VTE who are not on anticoagulant therapy. If a woman with prior VTE is receiving long-term anticoagulation, the use of COCs may be considered Category 3 (“risks generally outweigh benefits”) and should only be prescribed under hematologic and gynecologic supervision.

The recommended contraceptive options for women with a history of VTE include non-hormonal methods (copper IUD, barrier methods) or PBC, such as levonorgestrel-releasing intrauterine devices, progestogen-only pills, or subcutaneous progestogen implants [79]. It is advisable to avoid combined hormonal contraceptives and injectable progesterone in women with a history of thromboembolic events.

### 4.7. Family History of VTE and COC Use

To assess the risk of hormone-related thrombosis with a positive history of VTE in at least one parent or sibling, a Swedish nationwide case–control study analyzed 2311 cases of thrombosis in women aged 15–49 and did not find any significant links between family history of VTE and COC use (OR 0.92, 0.57–1.46) [80]. A reasonable explanation for this finding is the low prevalence of COC prescription among women with a family history of VTE. So, the authors suspect that they may underestimate the importance of a family history of VTE.

However, another Swedish study revealed that the risk of VTE associated with hormone use was doubled in women with a family history of VTE as compared to women with hormones and a negative family history [81].

Data from the START registry indicated that women of reproductive age are more prone to develop VTE in various situations, especially with COC use and pregnancy. They observed that cerebral vein thrombosis and pulmonary embolism occurred more frequently in the COC user group (14.8% and 36.9%, respectively) [82]. After adjusting for age, body mass index (BMI), smoking, hemoglobin level, cancer, thrombophilia, and family history of VTE, the researchers found that women with a family history had a significantly higher risk of VTE related to COC use than women with VTE who did not use COCs or who were pregnant.

### 4.8. Thrombophilia and the Risk of Thrombosis

Evaluation of genetic predisposition for thrombosis may identify an additional risk factor for venous thromboembolism development in COC users.

A meta-analysis of 12 case–control and three cohort studies performed in the Netherlands in the 2016 year showed that “mild” (FVL or prothrombin mutation) and “severe” (antithrombin, protein C and S deficiency, homozygosity or double heterozygosity for FVL or prothrombin mutation) thrombophilia increased the risk of VTE almost 6-fold (RR 5.89; 95% confidence interval [CI], 4.21–8.23) and 7-fold (RR, 7.15; 95% CI, 2.93–17.45), respectively, compared with COC users without genetic thrombophilia [83]. However, the authors of the meta-analysis believe that, in the absence of other risk factors, COCs can be offered to women with “mild” thrombophilic defects in situations where other reliable alternative contraceptives are not tolerated [83].

Martinelli et al.’s study found a strong link between oral contraceptive use and venous thrombosis, especially carriers of the prothrombin G20210A mutation [84] (OR 22.1 for cerebral vein thrombosis (CVT), 4.4 for deep vein thrombosis (DVT)).

Results from a prospective UK Biobank study assessing VTE risk factors related to OC use among 244,420 female participants showed that OC use was linked to an increased risk, especially in women with genetic thrombophilia [85]. They noted that women with high-risk genetic thrombophilia face a greater risk of thrombosis during the first two years of OC use (OR 6.35; 95% confidence interval, 4.98–8.09), with a general decline in risk over continued use. The highest risk was observed among carriers of the FV Leiden mutation (OR 5.73; 95% CI, 5.31–6.17) and the prothrombin gene mutation (OR 4.93; 95% CI, 3.16–7.7). Combining these two genetic variants significantly raises the risk of VTE to up to 15 times (OR 14.8; 95% CI, 9.28–23.6) [85].

Even though the risk of DVT is higher in adulthood, Krleza et al. described a case report of a seventeen-year-old girl with a history of OC use who developed DVT [86]. After a thorough examination, several genetic thrombophilic mutations were identified in this patient, including heterozygosity for factor V Leiden, prothrombin G20210A, and methylenetetrahydrofolate reductase (MTHFR) C677T gene mutations, as well as a homozygous 4G/4G genotype in the PAI-1 gene. Family history also revealed a few thromboembolic events. Thus, the presented clinical case confirms the multifactorial nature of the occurrence of deep vein thrombosis and allows us to conclude that there is a close relationship between the use of oral contraceptives in adolescents, a family history of thrombosis, and the development of thrombotic complications in the future.

The TREATS (Thrombosis: Risk and Economic Assessment of Thrombophilia Screening) study found that patients with FVL have a high risk of VTE while taking OCs (OR = 15.62; 95% CI 8.66–28.15) [87]. They showed that a multiplicative risk increase occurs when hereditary thrombophilia is combined with COC use for factor V Leiden or prothrombin mutation alone: approximately 4–5 times the risk; with COC use, up to a 30–35 times higher risk of VTE. A direct relationship between the development of VTE in patients taking OCs and deficiencies such as antithrombin (OR = 12.60; 95% CI 1.37–115.79), protein C (OR = 6.33; 95% CI 1.68–23.87), or protein S (OR = 4.88; 95% CI 1.39–17.10), as well as inherited increased factor VIII levels (OR = 8.80; 95% CI 4.13–18.75), was also identified.

Because of the high prevalence of FII G20210A and FV Leiden mutations in the Spanish population (6.5% and 2%, respectively) and the increased risk of VTE in COC users (3 times higher for the prothrombin mutation and 1.4 times higher for the FV mutation), the authors of another study recommend genetic screening in women from families with thrombophilia before prescribing COCs [88].

Dulicek et al., in their study of 770 patients, showed that hereditary thrombophilia, present in 44.5% of patients, is directly linked to the development of VTE in patients taking OCs [58]. In this study, the FVL mutation was the most common cause. Hereditary thrombophilia and antiphospholipid syndrome (APS) were diagnosed in nearly 45% of cases among women who develop OC-related issues and VTE. Reported COC use in APS is associated with markedly higher rates of venous and arterial thromboses [89]. COCs are absolutely contraindicated in women with definite APS (clinical + laboratory criteria), persistently positive lupus anticoagulant (LA), and multiple aPL positivity (LA + aCL + anti-β2GPI). Even asymptomatic carriers of antiphospholipid antibodies (aPL) have a higher baseline thrombosis risk compared with the general population. Safer alternatives are progestin-only pills (POP, desogestrel 75 µg Levonorgestrel IUS (LNG-IUS), copper IUD, barrier methods.

Currently, universal thrombophilia testing is not recommended when prescribing OCs. A simple activated protein C resistance test that detects the presence of the factor V Leiden mutation can provide a quick, affordable way to accurately identify and counsel women who are more prone to VTE before using OC, if a complete thrombophilic profile cannot be performed.

### 4.9. Smoking in COC Users

A study by Lidegaard in 1993 showed that the risk of cardiovascular diseases exceeds the risk of venous thromboembolic complications in women who smoke and use oral contraceptives [90]. Moreover, smoking and COC use have an additive and synergistic effect in increasing the risk of acute myocardial infarction and cerebral thrombosis (but not VTE), mainly with high doses of EE (50 mcg) [90].

Another extensive case–control study [91] showed that the combination of current smoking, oral contraceptive use, and FV Leiden mutation increased the risk of thrombosis by 5.0 times; for the FII G20210A mutation, this risk resulted in a 6.0-fold increase.

### 4.10. Obesity and Risk of Thrombosis

The evaluation of the effects of obesity and COC use on the risks of arterial and venous thrombosis identified that obese COC users (BMI) ≥30 kg/m^2^) had a higher risk for acute myocardial infarction [92], stroke [93], and cerebral venous thrombosis [94] compared with normal-weight nonusers. Regarding VTE, different prospective studies [95] found that obesity and COC use have a 5.0- to 8.0-fold increased risk of VTE compared with obese nonusers and a 10.0-fold increased risk compared with nonobese nonusers.

### 4.11. Arterial Hypertension

Estrogen-progestin drugs can increase BP by 5–7 mmHg, but persistent hypertension is very rare [96]. Prolonged OC use contributes to the occurrence of hypertension, a 1.9 (95% CI, 1.6–2.4) and 1.2 (95% CI, 1.1–1.5) times higher risk of hypertension during four-year follow-up among current and past OC users, respectively [97]. In this case, drospirenone, which eliminates the adverse effects of OCs on BP due to its anti-mineralocorticoid activity, can be recommended to patients for effective and safe contraception.

### 4.12. Dyslipidemia and COC Use

OCs may slightly raise triglyceride and HDL levels in the blood while lowering LDL levels. This mechanism also introduces some cardiovascular risks for patients, including the development of myocardial infarction or arterial thrombosis in other locations [98]. In women with dyslipidemia or metabolic syndrome, non-estrogen or low-estrogen contraception should be considered.

An alternative method of contraception in this case would be the use of a progestogen-only pill or the use of drospirenone [98].

### 4.13. Non-O Blood Type and the Risk of Thrombosis in COC Users

Multiple studies demonstrated that having a non-O blood type is considered an independent risk factor for VTE (OR, 1.98, 95% CI 1.57–2.49), (OR, 1.56, 95% CI, 1.43–1.69) [99,100,101] compared with those having the O blood group. It seems that ABO blood groups strongly affect the clearance of VWF. Due to increased VWF elimination, healthy individuals with type O have approximately 20% to 30% lower VWF levels compared to those in the non-O group [101], and according to a recent study, they have a higher susceptibility to ADAMTS13 proteolysis, which affects platelet function [102]. The addition of high-risk genetic thrombophilia to the non-O group increases this risk nearly threefold.

### 4.14. COVID/Post-COVID and the COC Use

Evidence that hormonal contraceptive users with COVID-19 are at heightened risk for venous or arterial thromboembolism is sparse and of low certainty [103]; no apparent increase in severe COVID outcomes in COC users, but COVID itself is a prothrombotic state, so additive risks must be considered.

### 4.15. Non-Steroidal Anti-Inflammatory Drugs and COC Use

The use of OCs in combination with other medications may also lead to venous thrombosis. Concomitant use of non-steroidal anti-inflammatory drugs (NSAIDs) and OCs was positively associated with an increased risk of DVT compared with a group of women who took OCs alone (OR = 7.2, 95% confidence interval 6.0–8.5) [104]. For this reason, women who simultaneously require NSAID therapy and hormonal contraception should be given informed counseling about the occurrence of possible risk factors.

### 4.16. Postpartum Period and COC Use

The risk of VTE is highest immediately after delivery and gradually declines. After 6 weeks (≥42 days) postpartum, in the absence of additional risk factors such as obesity, thrombophilia, cesarean delivery, maternal age over 35 years, or smoking, COCs may be started, provided the woman is not breastfeeding. However, individual VTE risk still needs to be evaluated [105].

### 4.17. Post-Oncologic Prothrombotic States

Women recovering from oncologic disease frequently exhibit prolonged prothrombotic states characterized by endothelial activation, elevated inflammatory cytokines, platelet hyperreactivity, and persistent coagulation abnormalities [106]. In such settings, the use of COCs is contraindicated or strongly discouraged, as estrogens may exacerbate the residual hypercoagulability and increase the risk of VTE. Progestin-only contraceptives (e.g., desogestrel-only pill, levonorgestrel IUD, or etonogestrel implant) and non-hormonal methods are considered safer alternatives [107].

Table 4 outlines major categories of women at increased thrombotic risk during COC use, together with recommended management strategies and safer contraceptive alternatives.

Women with a medical history of endogenous risk factors for venous thrombosis are in a separate category of patients at risk of thrombotic complications. Multiple studies indicate that the risk of thrombosis is increased in women with a personal history of VTE, gross obesity, age over 35 years, intimate partner violence [108], prolonged immobilization, in vitro fertilization [109], stress [110], hereditary thrombophilia, and antiphospholipid syndrome. However, none of these factors increases this risk more than pregnancy (5–20/10,000 woman-years) [111,112,113]; however, all these risks are additive [114]. Therefore, concomitant risk factors for venous thrombosis should be considered before prescribing OCs, and the patient’s family history of VTE and PE should be carefully assessed. Still, this decision also requires an individual approach to the patient. Modifiable risk factors for venous thrombosis, such as smoking and the use of prothrombotic drugs, should also be considered. To prevent the adverse effects of hormonal contraception, it is crucial to select the proper medication and dose. In this context, women at high risk of venous thrombosis might be advised to use microdosed oral agents.

## 5. Conclusions

Combined oral contraceptives are not uniform regarding thrombotic risk. The highest VTE risk is consistently linked to EE-containing COCs combined with third-generation progestins (desogestrel, gestodene) and cyproterone acetate. Preparations with levonorgestrel show a comparatively lower VTE risk and remain the preferred option for women without additional thrombotic risk factors. Recently introduced combinations with natural estrogens, especially estetrol/drospirenone, seem to have a more favorable hemostatic profile, although evidence is still emerging. An individual risk assessment—considering age, BMI, smoking, thrombophilia, and prior VTE—should guide the choice of COC and the necessity for follow-up.

## Figures and Tables

**Figure 1 ijms-26-11010-f001:**
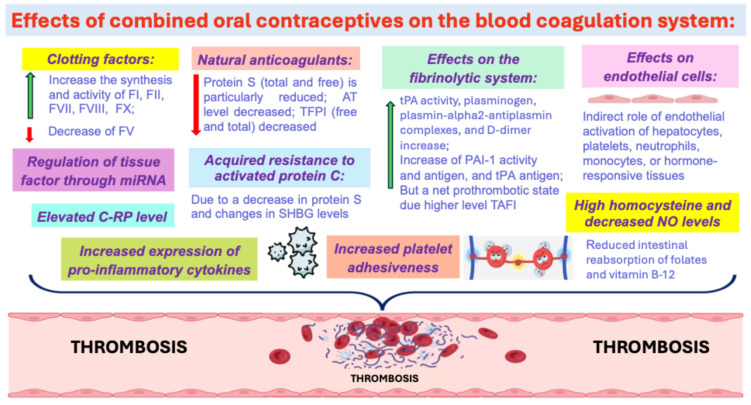
The thrombotic effects of COCs. Abbreviations: C-RP: C-reactive protein, TAFI—thrombin-activatable fibrinolysis inhibitor, SHBG—sex hormone-binding globulin, TFPI—tissue factor pathway inhibitor.

**Table 1 ijms-26-11010-t001:** Progestogens characteristics in COCs.

Generation	Type of Progestogen
First	Cyproterone acetate	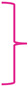	
Medroxyprogesterone acetate	Pregnanes (carbon-21)
Chlormadinone acetate	from 17-OH Progesterone
Norethynodrel, Norethindrone,	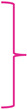	
Ethynodiol diacetate, Lynestrol	Estranes (carbon-18)
Norethisterone acetate (also known as norethindrone acetate)	from Testosterone
Second	Levonorgestrel (LNG)	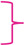	Gonanes (carbon-17)
Norgestrel	from Testosterone
Third	Gestodene	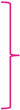	
Etonorgestrel	Gonanes from
Desogestrel	Levonorgestrel
Norgestimate/Norelgestromine	
Forth	Dienogest (DNG)	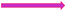	gonane derivative
Drospirenone (DRSP)	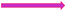	derivate from spironolactone
Nestorone (NES)	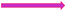	19-norprogesterone derivative
Nomegestrol acetate (NOMAc)	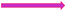	from norpregnane (carbon-20)
Trimegestone (TMG)	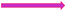	19-norprogesterone derivative

**Table 2 ijms-26-11010-t002:** Pathogenetic mechanisms involved in COC-related thrombosis.

Mechanism/System	Effect of COCs	Representative Mediators/Notes
Activation of coagulation cascade	Increased synthesis of procoagulant factors (FII, FVII, FVIII, FX, fibrinogen) leading to hypercoagulability	Estrogen-dependent; most pronounced with third-generation progestins
Reduction in natural anticoagulants	Decreased protein S and antithrombin, acquired APC resistance	Degree of APC resistance correlates with estrogen dose and SHBG levels
Decrease in TFPI	Reduction in total/free TFPI levels, contributing to mild prothrombotic imbalance	Weak but additive effect, particularly relevant in thrombophilia
Endothelial dysfunction	Indirect activation, reduced nitric oxide (NO) bioavailability, oxidative stress	Mediated by inflammation and reactive oxygen species
Impairment of fibrinolysis	Increased TAFI levels counteract the rise in tPA/plasmin activity	Net antifibrinolytic effect; stronger with desogestrel-containing COCs
Platelet activation	Enhanced aggregation and adhesion, increased P-selectin	Driven by estrogen–platelet interactions
Inflammation	Elevated CRP, IL-6, and NF-κB activity	Reflects systemic proinflammatory milieu
Homocysteine–NO axis	Reduced folate/B12 absorption → ↑ homocysteine → ↓ NO synthesis	Contributes to endothelial dysfunction
microRNAs	Dysregulation of miR-365a-3p and miR-494-3p alters TF and protein S expression	Potential epigenetic mechanism
Neutrophil extracellular traps (NETs)	Release of chromatin–protein complexes promoting coagulation	Both biomarker and therapeutic target candidate
Lipoprotein(a) [Lp(a)]	Antifibrinolytic and proinflammatory properties; possible enhancement of COC-related thrombotic risk	Requires further clinical validation

**Table 3 ijms-26-11010-t003:** Comparative venous thromboembolism (VTE) risk among COC formulations.

Estrogen/Progestin Combination	Representative Product	Relative VTE Risk vs. Non-Users	Comments
EE 30–35 µg + levonorgestrel (2nd generation)	30 EE/LNG	3–4× baseline	Reference low-risk COC
EE 30–35 µg + desogestrel or gestodene (3rd generation)	EE/DSG, EE/GSD	5–7×	Higher risk than EE/LNG
EE + drospirenone	EE/DRSP	5–6×	~1.5-fold higher than EE/LNG
EE + cyproterone acetate	EE/CPA	6–8×	Among the highest risks
Low-dose EE (20 µg) + LNG	20 EE/LNG	2–3×	Slightly reduced risk vs. 30 µg EE
Estradiol valerate + dienogest	E2V/DNG	≤LNG level	“Safer alternative”
17β-Estradiol + nomegestrol acetate	E2/NOMAC	Lower than EE-based COCs	Favorable hemostatic profile
Estetrol (E4) + drospirenone	E4/DRSP	Presumably ↓ vs. EE/DRSP	Emerging evidence of lower VTE risk
Progestin-only contraceptives	POP, LNG-IUS	≈background	Recommended for women with thrombophilia or VTE history

Abbreviations: CPA—cyproterone acetate; DNG—dienogest; DRSP—drospirenone; DSG—desogestrel; EE—ethinylestradiol; E2V—estradiol valerate; E2—17β-Estradiol; E4—Estetrol; GSD—gestodene; LNG—levonorgestrel; LNG-IUS—levonorgestrel-releasing intrauterine system; NOMAC—nomegestrol acetate; POP—progestin-only contraceptives.

**Table 4 ijms-26-11010-t004:** Women at increased thrombotic risk during COC use and recommended management strategies.

Category of Women	Mechanisms Increasing Thrombotic Risk	Preferred Contraceptive Options/Management Considerations	Approx. VTE Risk vs. Healthy Non-User
Hereditary thrombophilia (FVL, Prothrombin G20210A, antithrombin, protein C/S deficiency)	Additive effect with COC-induced APC resistance; reduced natural anticoagulants	Avoid estrogen COCs; prefer POP, implant, LNG-IUS; individualized counseling	High–very high. Heterozygous FVL + COC often quoted ~10–30× [87]; prothrombin G20210A + COC ~5–15× [84]
Non-O blood type (A/B/AB)	Higher vWF & FVIII → increased baseline coagulability; synergy with estrogen	Include blood group in risk stratification; if other risks present → prefer POP/LARC or lowest-dose EE	Low–moderate alone (~1.5–2×); additive with COC (roughly 2–4× total) [99,100,101,102]
Elevated lipoprotein(a)	Antifibrinolytic, pro-inflammatory; endothelial dysfunction	Check Lp(a) with family history of VTE/early CVD; prefer low-estrogen or non-estrogen methods	Low–moderate (~1.5–2×); may rise with COC [51,52]
Obesity (BMI ≥30 kg/m^2^)	Chronic inflammation, venous stasis, estrogen storage in adipose tissue	Low-dose EE or progestin-only; counsel weight control & activity	Moderate alone (~2–3×); with COC can reach 5–10× [92,93,94,95]
Smoking (>35 years)	Endothelial injury, platelet activation → mainly arterial risk	Strongly discourage COCs; consider POP/implant/LNG-IUS	Primarily arterial risk (MI/Stroke ↑); VTE slight–moderate ↑ [90]
Antiphospholipid syndrome (APS)	Persistent hypercoagulability (platelet/endothelial activation)	COCs contraindicated; use progestin-only or non-hormonal	Very high baseline; estrogen adds unacceptable risk
History of VTE	Prothrombotic milieu; impaired fibrinolysis	Avoid estrogen; use POP or non-hormonal (Cu-IUD, barriers)	High recurrence risk; COC adds 2–4× or more [53]
Hypertension/dyslipidemia	Endothelial dysfunction, oxidative stress	Consider low-dose/natural-estrogen with close monitoring, or switch to POP	Low–moderate for VTE; arterial risk predominates [96,97,98]
Post-oncologic or post-COVID prothrombotic states	Residual inflammation; endothelial/platelet activation	Defer/avoid COCs until labs normalize; prefer non-estrogen temporarily	Variable; often moderate–high until recovery [103,106,107]
Postpartum period	Physiologic hypercoagulability & venous stasis to ~6 weeks	Defer COCs until ≥6 weeks; POP or LARC preferred	Very high baseline (~20–60×); avoid estrogen early [105]

## Data Availability

No new data were created or analyzed in this study. Data sharing is not applicable to this article.

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
