# Peer review of "Combined Oral Contraceptives and the Risk of Thrombosis"

_ijms, 2025, doi:10.3390/ijms262211010_

Round 1

Reviewer 1 Report

Comments and Suggestions for Authors

After reading the article I have the following comments:

  • the manuscript is more like a book chapter than a narrative review.
  • it has to be mentioned clearly in the introduction both the type of the article and its aim.
  • the methodology of the study is not described (which databases have been searched, what MeSH terms have been used, the time period etc).
  • the article focuses on thrombosis in COCs users from mechanisms to risk as mentioned in the title but the last part also presents the impact of cardiovascular risk factors (smoking, obesity, dyslipidemia) on COCs users and even the definition of arterial hypertension and the effect COCs on blood pressure. This information beyond the scope of the article.
  • newer COCs like estetrol/drospirenone which seems to have a lower VTE risk compared to other combinations containing drospirenone should be discussed.
  • the conclusion must reflect the key messages of the research (for example, which COC has the highest risk of thrombosis and which has the lowest). In the present form it contains recommendations and general truths like "reducing the risk improves women's quality of life" although the research does not focus on quality of life!

Author Response

We want to thank the Reviewer for taking the time to review this manuscript and for providing valuable feedback. We have revised the article in accordance with the suggestions provided. Please find below a detailed point-by-point response. All corrections are highlighted in the re-submitted file.

Point-by-point response to Comments and Suggestions for Authors

Comments 1: [The manuscript is more like a book chapter than a narrative review]

Response 1:  

Thank you for this remark. The structure of the manuscript has been revised to better align with the format of a narrative review. Redundant background sections were shortened, and the focus was redirected toward the synthesis and interpretation of published evidence. The new sections “Methodology” and “Conclusion” were reorganized to highlight the main findings rather than descriptive content.

(Revised in Introduction, pp. 2–3.)

Comments 2: [It has to be mentioned clearly in the introduction both the type of the article and its aim]

Response 2: 

Thank you for this valuable suggestion. The Introduction now explicitly defines the paper as a narrative review and states its objective. (Added in Introduction, p. 2)

‘This article is a narrative review aimed at summarizing the current evidence on the mechanisms and risk of thrombosis associated with combined oral contraceptive (COC) use, emphasizing differences between formulations and identifying the safest hormonal combinations in terms of thrombotic risk.’

Comments 3: [The methodology of the study is not described (which databases have been searched, what MeSH terms have been used, the time period etc).]

Response 3: 

Thank you for this comment. We completely agree and have added a new subsection titled “Methodology” to the Introduction. It now specifies the databases (PubMed, Scopus, Web of Science), key MeSH terms (e.g., “combined oral contraceptives,” “thrombosis,” “venous thromboembolism,” “risk factors”), and the time period (January 2010–June 2024) used for the literature search (New Section 2, p. 4.).

“2. Methodology”

“A structured literature search was performed across PubMed, Scopus, and Web of Science databases to find studies published from January 2010 to June 2024. Various combinations of MeSH terms and keywords were used, including: “combined oral contraceptives,” “venous thromboembolism,” “thrombosis,” “risk factors,” “progestins,” and “estetrol drospirenone.” Preference was given to systematic reviews, meta-analyses, and clinical studies written in English. Reference lists of relevant publications were also screened to find additional sources.”

Comments 4: [The article focuses on thrombosis in COCs users from mechanisms to risk as mentioned in the title but the last part also presents the impact of cardiovascular risk factors (smoking, obesity, dyslipidemia) and even the definition of hypertension. This information is beyond the scope of the article.]

Response 4: 

We appreciate this observation. The section discussing cardiovascular risk factors and hypertension definitions has been condensed and partially removed. Only information directly relevant to thrombosis risk in COC users has been retained, ensuring the manuscript remains focused on the main topic (Revised in Section “Risk Factors for Thrombosis,” pз. 17-19)

Comments 5: [Newer COCs like estetrol/drospirenone which seem to have a lower VTE risk compared to other combinations containing drospirenone should be discussed.]

Response 5: 

Thank you for highlighting this critical point. A new paragraph discussing estetrol/drospirenone formulations has been added to the section ‘Composition of Oral Contraceptives and Risk of Thrombosis.’ (Added on p. 13.)

“In recent years, a new combined oral contraceptive containing estetrol (E4) and drospirenone has been introduced. Estetrol is a natural fetal estrogen with selective action on estrogen receptors and has a weaker effect on hepatic synthesis of coagulation factors compared to ethinylestradiol. Early clinical data and postmarketing observations suggest that E4/drospirenone may have a more favorable hemostatic profile and potentially a lower VTE risk than EE-containing preparations with drospirenone, while still offering effective contraception. However, these findings still require confirmation in larger real-world studies cohorts.”

Comments 6: [The conclusion must reflect the key messages of the research (for example, which COC has the highest risk of thrombosis and which has the lowest). In the present form it contains general statements.]

Response 6: 

We have revised the conclusion to briefly summarize the key findings, emphasizing the relative thrombotic risks of different COC formulations (e.g., ethinylestradiol + desogestrel > ethinylestradiol + levonorgestrel > estetrol + drospirenone). General statements about quality of life have been removed. (p. 21).

“Combined oral contraceptives are not uniform regarding thrombotic risk. The highest VTE risk is consistently linked to EE-containing COCs combined with third-generation progestins (desogestrel, gestodene) and cyproterone acetate. Preparations with levonorgestrel show a comparatively lower VTE risk and remain the preferred option for women without additional thrombotic risk factors. Recently introduced combinations with natural estrogens, especially estetrol/drospirenone, seem to have a more favorable hemostatic profile, although evidence is still emerging. An individual risk assessment—considering age, BMI, smoking, thrombophilia, and prior VTE—should guide the choice of COC and the necessity for follow-up.”

Reviewer 2 Report

Comments and Suggestions for Authors

Excellent review! Congratulations!

The aim of this article is to review the mechanisms that lead to the onset of both venous and arterial thrombotic events in women taking COCs. All the mechanisms that lead to clot formation through the prothrombotic imbalance of all components of the hemostatic system (endothelium, platelets, coagulation, fibrinolysis) are illustrated in detail. Other mechanisms that increase thrombotic risk are also listed: inflammation, increased homocysteine levels, decreased NO, microRNA activity, and the appearance of NETs. After a brief historical overview, the current classification of COCs (from first to fourth generation) is recalled, and the varying levels of thrombotic risk associated with the various drugs of the four generations is highlighted. It highlights the categories of women at greatest risk of thrombosis and how thrombotic risk evolves over time and therefore must be constantly monitored. In my opinion, the topic of the review is always extremely timely and is a very useful "review" of many issues that are not always given careful consideration (for example, women with thrombophilia and what type of thrombophilia they have). The article does not fill specific gaps in the field, but it has the great advantage, as a review, of condensing all the issues related to the topic covered in a single article.

In the context of thrombophilias, however, if you included TFPI, which has a weak thrombogenic effect, it would have been interesting to also include LP(a), a parameter currently under extensive research.

The single figure present could certainly have been nicer, but it includes all the mechanisms involved in the pathogenesis of thrombosis. Furthermore, I would have liked some tables, which I find very useful in a summary.
In summary, in my opinion, this review is a very useful summary tool on pathogenic mechanisms of COCs, on the categories of women at increased thrombotic risk, on the need for constant monitoring, and it also highlights more recent acquisition mechanisms such as homocysteine, NO, and especially NETs, with a view not only as a biomarker but also as a therapeutic target.

For the rest, everything is fine except the bibliography, because there are two groups of bibliographic citations: which should be kept and which should be excluded?

Author Response

We sincerely thank Reviewer 2 for the positive and encouraging evaluation of our manuscript and for the thoughtful suggestions that helped us further improve its clarity and completeness. We have carefully considered each point and revised the manuscript accordingly.

Point-by-point response to Comments and Suggestions for Authors

Comments 1: [In the context of thrombophilias, however, if you included TFPI, which has a weak thrombogenic effect, it would have been interesting to also include LP(a), a parameter currently under extensive research.]

Response 1:  

Thank you for this valuable suggestion. We have added a brief paragraph discussing the role of lipoprotein(a) [Lp(a)] as an emerging thrombogenic factor, particularly relevant in women using combined oral contraceptives and in those with hereditary thrombophilia. The new text appears in the section “Thrombophilias and prothrombotic markers” (p. ):

“Recent studies have also highlighted the potential role of lipoprotein(a) (Lp(a)) as a prothrombotic factor in the context of COC use. Elevated plasma levels of Lp(a) may promote both venous and arterial thrombosis through different pathways, including antifibrinolytic and proinflammatory mechanisms, such as inhibiting plasminogen activation and increasing oxidative stress. Oxidized phospholipids carried by Lp(a), along with vascular endothelial dysfunction, may further raise the thrombotic risk associated with COC use. Although the clinical significance of Lp(a) in women using COCs remains limited, measuring it could provide additional insights for women with hereditary or acquired thrombophilia, supporting a more personalized assessment of thrombotic risk.”

Comments 2: [The single figure present could certainly have been nicer, but it includes all the mechanisms involved in the pathogenesis of thrombosis.]

Response 2: 

We appreciate this helpful comment. The figure has been slightly adjusted for improved readability and visual balance.

Comments 3: [I would have liked some tables, which I find very useful in a summary.]

Response 3: 

We thank the Reviewer for this valuable suggestion. In response, we have added a new summary table (Table 2) that presents the primary mechanisms leading to venous and arterial thrombosis in COC users, a second table (Table 3) comparing the relative thrombotic risks among different COC generations and formulations, and a third new table (Table 4 in the text) outlining major categories of women at increased thrombotic risk during COC use, along with recommended management strategies and safer contraceptive alternatives.

These tables are located on pp.9-10, 14, and 19-20, respectively.

Comments 4: [For the rest, everything is fine except the bibliography, because there are two groups of bibliographic citations: which should be kept and which should be excluded?]

Response 4: 

The bibliography has been carefully revised and unified. All duplicated or obsolete references have been removed, and a few new ones have been added.

Round 2

Reviewer 1 Report

Comments and Suggestions for Authors

My comments have been addressed. The manuscript has improved significantly.